# Combined L-Citrulline Supplementation and Slow Velocity Low-Intensity Resistance Training Improves Leg Endothelial Function, Lean Mass, and Strength in Hypertensive Postmenopausal Women

**DOI:** 10.3390/nu15010074

**Published:** 2022-12-23

**Authors:** Yejin Kang, Katherine N. Dillon, Mauricio A. Martinez, Arun Maharaj, Stephen M. Fischer, Arturo Figueroa

**Affiliations:** 1Department of Kinesiology and Sport Management, Texas Tech University, Lubbock, TX 79409, USA; 2Department of Epidemiology and Cancer Control, St. Jude Children’s Research Hospital, Memphis, TN 38105, USA

**Keywords:** citrulline, resistance training, endothelial function, arterial stiffness, muscle mass, muscle strength, postmenopausal women, hypertension

## Abstract

Hypertension is highly prevalent in postmenopausal women. Endothelial dysfunction is associated with hypertension and the age-related decreases in muscle mass and strength. L-citrulline supplementation (CIT) and slow velocity low-intensity resistance training (SVLIRT) have improved vascular function, but their effect on muscle mass is unclear. We investigated whether combined CIT and SVLIRT (CIT + SVLIRT) would have additional benefits on leg endothelial function (superficial femoral artery flow-mediated dilation (sfemFMD)), lean mass (LM), and strength in hypertensive postmenopausal women. Participants were randomized to CIT (10 g/day, *n* = 13) or placebo (PL, *n* = 11) alone for 4 weeks and CIT + SVLIRT or PL + SVLIRT for another 4 weeks. sfemFMD, leg LM and muscle strength were measured at 0, 4, and 8 weeks. CIT increased sfemFMD after 4 weeks (CIT: Δ1.8 ± 0.3% vs. PL: Δ−0.2 ± 0.5%, *p* < 0.05) and 8 weeks (CIT + SVLIRT: Δ2.7 ± 0.5% vs. PL + SVLIRT: Δ−0.02 ± 0.5, *p* = 0.003). Leg LM improved after CIT + SVLIRT compared to PL + SVLIRT (Δ0.49 ± 0.15 kg vs. Δ0.07 ± 0.12 kg, *p* < 0.05). Leg curl strength increased greater with CIT + SVLIRT compared to PL + SVLIRT (Δ6.9 ± 0.9 kg vs. Δ4.0 ± 1.0 kg, *p* < 0.05). CIT supplementation alone improved leg endothelial function and when combined with SVLIRT has additive benefits on leg LM and curl strength in hypertensive postmenopausal women.

## 1. Introduction

Hypertension is the main risk factor for cardiovascular disease (CVD), the leading cause of death in American women [1]. Postmenopausal women are at high risk for developing heart failure with preserved ejection fraction due to increased aortic stiffness and systolic blood pressure (SBP) [2]. Arterial stiffening and systolic hypertension are preceded by impaired endothelial function characterized by reduced nitric oxide (NO) bioavailability [3]. Low availability of L-arginine (ARG), the substrate for NO synthesis, contributes to endothelial dysfunction in postmenopausal women [4]. Growing evidence suggests that endothelial dysfunction and high arterial stiffness are associated with decreased limb lean mass (LM) and strength in older adults, particularly in women [5,6,7,8]. The impact of arterial stiffening is evident on the thigh muscles of older women [9]. In fact, evidence suggests that age-related limb vascular dysfunction may impair muscle blood flow and contribute to declines in muscle mass and strength [8,10]. Inverse associations between vascular dysfunction and skeletal muscle abnormalities are observed in hypertensives [10]. Therefore, nutritional and exercise interventions are suggested to improve vascular and muscular function in the hypertensive population.

Dietary supplements to augment plasma ARG levels have shown effectiveness for improving vascular function. Although short-term ARG supplementation has improved endothelial function in older adults [11], its efficiency following long-term periods declines due to ARG catabolism by increased arginase activity [12]. L-citrulline (CIT) bypasses arginase catabolism and liver extraction and inhibits arginase activity [13]. Thus, oral CIT leads to greater plasma ARG levels than a similar dose of ARG [14]. Oral CIT supplementation has improved brachial artery endothelial function (flow-mediated dilation, FMD) in patients with vasospastic angina [15] and decreased leg arterial stiffness (femoral–ankle pulse wave velocity, faPWV) in hypertensive postmenopausal women [16]. Although brachial FMD is the gold-standard measure of endothelial function, it is not associated with superficial femoral FMD (sfemFMD) [17]. The impact of CIT supplementation on sfemFMD should be explored since leg arteries and muscles are more susceptible to the deleterious effects of aging and disease [17,18,19].

Growing evidence supports an anabolic action of CIT supplementation in skeletal muscle protein in adults with low-protein consumption [20]. Moreover, 3 weeks of CIT supplementation increased total body LM in malnourished older adults [21,22] by enhancing muscle protein synthesis [21]. Devries et al. [23] investigated the impact of CIT supplementation (5 g/day) for 2 weeks on physical inactivity-induced decreases in muscle mass in older men. The ineffectiveness of CIT for increasing muscle mass may be related to the short duration of the intervention. The muscle anabolic effect of CIT supplementation may not be evident in women without nutritional deficiencies [16]. It is also possible that 10 g/day of CIT would be an effective dose for muscle improvements [21,24].

Previous literature suggests 12 weeks of both aerobic and resistance training (RT) have similar effects on brachial FMD in middle-aged adults with hypertension [25]. However, implementing RT to augment muscle mass and strength may be an important exercise modality to improve endothelial function and muscle properties. The feasibility of postmenopausal women to perform high-intensity RT is low due to exaggerated SBP responses to acute exercise and injury risk [26,27]. Alternatively, slow velocity low-intensity resistance training (SVLIRT) improved muscle mass and strength similarly to high-intensity RT in young men [28]. Furthermore, SVLIRT for 8 to 12 weeks improved brachial FMD and systemic arterial stiffness in young men [29]. However, older adults have reduced capacity to increase muscle mass, but not strength, in response to leg RT due to anabolic resistance [30]. Previous studies found improvements in vascular function and muscle strength, but not leg LM, after 12 weeks of SVLIRT in postmenopausal women [31,32], suggesting a blunted muscle anabolic response.

There is an age-related decline in leg LM in women 45 years and older [19], suggesting an increased need for nutritive blood flow to the lower extremities. Since muscle mass and strength are positively associated with endothelial function [5,6], CIT supplementation combined with exercise training would improve LM via NO production [33]. CIT supplementation (6 g/day) combined with whole-body vibration training (WBVT), a type of strength exercise, for 8 weeks improved aortic stiffness and leg LM compared with CIT or training alone in obese hypertensive postmenopausal women [16]. Moreover, Buckinx and colleagues [24] reported improvements in handgrip strength following high-intensity interval training combined with CIT supplementation (10 g/day) for 12 weeks compared with training alone in obese dynapenic older adults.

The purpose of this study was to investigate the effects of 4-weeks of CIT supplementation alone followed by 4-weeks of combined CIT supplementation with SVLIRT on leg vascular function (sfemFMD and faPWV), LM, and muscle strength in hypertensive postmenopausal women. We hypothesized that CIT supplementation alone would improve sfemFMD and faPWV without muscular benefits compared to placebo (PL). Moreover, we hypothesized that CIT supplementation plus SVLIRT would increase sfemFMD, faPWV, leg LM and muscle strength while PL would increase only muscle strength in postmenopausal women with hypertension.

## 2. Materials and Methods

### 2.1. Participants

Twenty-four postmenopausal women (≥1 year without menstruation) aged 50 to 75 years were recruited from the local communities via social media advertisements. All participants were sedentary women (<120 min/week of physical activity or exercise at low to moderate intensity during the past 6 months) with a body mass index (BMI) ≤ 40 kg/m^2^ and a resting SBP between 120 and 159 mmHg for medicated hypertensives and between 130 and 159 mmHg for unmedicated individuals. Participants were excluded if diagnosed with cardiovascular disease, diabetes (type 1 or 2), and any renal/pulmonary/metabolic disorders or if they were taking any beta-blockers, anti-inflammatory drugs, or vasoactive supplementations. Participants were also excluded if they had any orthopedic limitations interfering with their ability to perform the exercises in the study.

### 2.2. Study Protocol

This was a randomized, double-blind, PL-controlled, parallel-design study. All pre-qualified participants following an initial telephone screening visited the Vascular Health Lab at Texas Tech University. Before each visit, participants were asked to abstain overnight from food, medications, and caffeine for at least 12 h and refrain from alcohol for at least 24 h. The study protocol was approved by the Institutional Human Subject Committee of Texas Tech University (IRB2018-463) and conformed to the standards set by the Declaration of Helsinki. The study was registered in ClinicalTrials.gov under NCT05227781.

After collecting measurements of vascular function, body composition and strength at 0 week, participants were randomized to either CIT supplementation or PL alone for 4 weeks. The same measurements collected a 0 week were collected at 4 and 8 weeks. Following the 4th week visit, participants were instructed to continue with the same previously assigned supplementation and started SVLIRT for an additional 4 weeks. Figure 1 shows the study flow chart. We analyzed and reported data from participants who completed the 8 weeks of intervention. Figure 2 presents the experimental protocol.

### 2.3. Anthropometrics and Body Composition

Anthropometry and body composition were measured during the first visit after participants signed an informed consent form and completed a health and exercise history questionnaire. Height and weight were measured using a stadiometer (Free-Standing Portable Height Rod, Detecto, Webb City, MO, USA) and a beam scale (Weigh Beam, Detecto, Webb City, MO, USA). BMI was calculated as weight (kg) divided by height squared (m^2^). Waist circumference was measured using a non-elastic tape measure at the midpoint between the superior border of the iliac crest and the lower border of the last rib. Visceral adipose tissue (VAT), legs LM, arms LM, and total percent fat were assessed by a whole-body dual-energy X-ray absorptiometry (DEXA) scan (GE Lunar DPX-IQ, Madison, WI, USA).

### 2.4. Measurement of Resting BP and faPWV

Cardiovascular measures were conducted between 6 and 10 AM in a quiet, dark, temperature-controlled room (~23 °C). After 20 min of rest in the supine position, brachial blood pressure (BP) was obtained in the non-dominant arm using an automated oscillometric device (HEM-907XL; Omron Healthcare, Vernon Hill, IL, USA) and averaged from at least two readings with a <5 mmHg difference. FaPWV was measured using a tonometer placed on the common femoral artery and ankle (dorsalis pedis artery) and a 3-lead electrocardiography (ECG) connected with the device (SphygmoCor CPV; AtCor Medical, Sydney, Australia). The distance between the femoral and ankle points was measured using a segmometer (Veriner Caliper; Mitutoyo, Kawasaki, Japan). FaPWV was estimated by capturing 10–15 consecutive pulse waves from the two arteries and calculating the distance divided by the pulse transit time. Pulse transit time was calculated from the R-wave of the ECG to the feet of the captured pulse waves. At least two PWV measurements were taken and used for data analysis, ensuring a ≤0.3 m/s difference between both readings.

### 2.5. Measurement of sfemFMD

A rapid-inflating cuff (Hokanson E20, Bellevue, WA, USA) was placed proximal to the knee joint and a 12-MHz linear array Doppler ultrasound probe (Logiq S7; GE Healthcare, Milwaukee, WI, USA) was positioned on the superficial femoral artery using a probe holder. After a 2 min baseline measurement, the cuff was rapidly inflated to 250 mmHg for 5 min to completely occlude the artery. Following 5 min of ischemia, the cuff was rapidly deflated, and the artery diameter and mean blood velocity were recorded during 3 min of reactive hyperemia. The entire 10 min video was recorded using an open-source video recording software (OBS Studio), and the recording was analyzed using an edge-detection software (Cardiovascular Suite, Quipu, Pisa, Italy). SfemFMD was calculated as: FMD (%) = (peak diameter−baseline diameter)/baseline diameter × 100.

### 2.6. Measurement of Leg Muscle Strength

Leg muscle strength was assessed using 10-repetition maximum (10RM) testing for the leg press, leg extension, leg curl, and calf raise exercises at 0, 4 and 8 weeks using the same criteria. The 1RM may have elicited exaggerated rises in BP in this hypertensive population and was avoided for safety concerns. The 10RM, the maximum amount of weight lifted 10 times through complete range of motion, was obtained within 3–4 sets with 90 s of rest between sets. A 10RM was achieved when participants failed to proceed due to muscle fatigue. The 1RM was calculated to determine the intensity of training using the validated Brzycki equation [34]: Estimated 1RM = weight lifted/(102.78−(2.78 × repetitions)).

### 2.7. CIT Supplementation and SVLIRT

Participants were randomized to either CIT or PL groups by a researcher not involved in data collection. Participants were asked to consume a daily dose of 10 g of CIT or PL (maltodextrin) by taking 6 pills in the morning and 7 at night for 4 weeks (CIT and PL provided by NOW Foods, Bloomingdale, IL). PL was provided in capsules with similar color, size, and weight as the CIT capsules. After the 4-week lab visit, all participants were re-supplied with the same supplement and began SVLIRT for 4 weeks.

SVLIRT consisted of 4 lower body exercises (leg press, leg extension, leg curl, and calf raise) lasting approximately 25 min per session, 3 times a week, for 4 weeks. The intensity of training was set at 40% and 50% of estimated 1RM for the first and second 2 weeks, respectively. Participants performed all exercise movements with a slow speed contraction (3 s concentric and 3 s eccentric) using a metronome for 3 sets of 15 repetitions with 1–3 min of rest between sets. All training sessions were supervised by a properly trained graduate student, and the rate of perceived exertion (RPE) was recorded after every set.

To ensure adherence to supplements, bi-weekly phone calls were conducted by an investigator. Participants were asked to bring their supplement bottles back to the laboratory at the 4- and 8-week visits, and capsules were counted to calculate compliance to supplementations. Retention and compliance for the SVLIRT were monitored through training logs.

### 2.8. Statistical Analysis

To determine sample size, power analysis was conducted using G*Power (version 3.1.9.4, Dusseldorf, Germany). Based on previous study [11], it was estimated that 8 participants per group would improve brachial FMD by 3.06% with a power of 86% at α = 0.05 level. Normal distribution of data was confirmed using the Shapiro–Wilk test. Independent *t*-test was used to compare baseline differences between groups. To determine differences between groups (CIT vs. PL) over time (0-, 4-, and 8-week), a two-way repeated measures analysis of variance with Bonferroni adjustments was performed. When a significant group-by-time interaction was identified, post hoc comparisons were performed via *t*-tests. Pearson’s correlations were used to examine relationships between the changes in leg LM and total percent fat with the change in sfemFMD from 4 to 8 weeks. All statistical analyses were conducted using SPSS 26.0 (IBM SPSS Statistics, Chicago, IL, USA). Data were presented as mean ± standard error (SE). *p* < 0.05 was considered as statistical significance.

## 3. Results

Participant characteristics and medication are shown in Table 1. There were no significant baseline differences between two groups (all *p* > 0.05). Compliance to supplementation was 92 ± 7% (0 week to 4 weeks) and 96 ± 6 % (4 weeks to 8 weeks). All participants completed 95 ± 7% of the training sessions. There were no adverse effects experienced by the participants during the study.

SfemFMD and leg arterial stiffness are showed in Table 2. There were significant group-by-time interactions for sfemFMD (*p* < 0.001). Figure 3 shows that sfemFMD significantly increased after 4 weeks of CIT supplementation compared to 0 week (*p* < 0.001, Figure 3A) and PL (*p* < 0.05, Figure 3B). The addition of SVLIRT to CIT supplementation tended to increase sfemFMD (*p* = 0.093) from 4 to 8 weeks. Changes in sfemFMD from 0 week to 4 weeks (*p* < 0.05) and 8 weeks (*p* < 0.001) were significantly different between CIT and PL groups (Figure 3B). CIT and PL did not affect faPWV after 4 weeks, but both groups had similar significant reductions in faPWV after the combination with SVLIRT (*p* < 0.05, Figure 4).

Body composition and leg strength are reported in Table 3. At 0 weeks, there were no differences in body composition or indices of leg strength between the CIT and PL groups (all *p* > 0.05). Significant group-by-time interactions were found for leg LM and leg curl 10RM (all *p* < 0.05, Table 3). No significant improvement was observed in leg LM after 4 weeks of PL and CIT alone. Leg LM significantly increased after 8 weeks of CIT and 4 weeks of SVLIRT compared to PL (*p* < 0.05, Figure 5A,B). The changes in leg LM from 0 to 4 (*p* < 0.05), 0 to 8 (*p* < 0.001), and 4 to 8 (*p* < 0.05) weeks were significant in the CIT group compared to PL (Figure 5B). Although leg curl strength significantly improved after CIT supplementation alone (*p* < 0.05, Figure 6), the increase was no different than PL. Leg strength (leg press, leg extension, and calf raises) did not increase after 4 weeks of CIT and PL alone. Both groups experienced increases in leg press, leg extension, leg curl, and calf raise strength after 4 weeks of SVLIRT (all *p* < 0.05, Table 3). However, 8 weeks of CIT supplementation increased leg curl strength more than PL (*p* < 0.05, Figure 6A,B).

Figure 7 shows correlations between changes in sfemFMD, leg LM, and total percent fat from 4 weeks to 8 weeks. The change in sfemFMD was correlated with change in leg LM (r = 0.436, *p* < 0.05, Figure 7A) and total percent fat (r = −0.528, *p* < 0.01, Figure 7B).

## 4. Discussion

Our findings show that 4 weeks of CIT supplementation alone improved leg endothelial function (sfemFMD) and leg curl strength. The addition of SVLIRT to CIT supplementation for another 4 weeks further enhanced sfemFMD and resulted in greater improvements in leg LM and leg curl strength compared to PL. Although 8 weeks of CIT supplementation did not affect leg arterial stiffness (faPWV), SVLIRT reduced faPWV and increased leg strength (other than curl) similarly in both groups. Our findings suggest that CIT supplementation effectively increased leg LM and curl strength via improvements in leg endothelial function when combined with SVLIRT in hypertensive postmenopausal women.

In postmenopausal women, low ARG availability for NO production impairs endothelial function [4], which is associated with decreased limb LM and strength [5,6]. Endothelial dysfunction is commonly assessed as low brachial FMD, a recognized predictor of CVDrisk [35,36]. However, brachial FMD does not reflect leg endothelial function [17]. Vascular impairments with aging are limb specific [18] and leg arteries have a higher incidence of vascular dysfunction and disease than the brachial artery [17]. Moreover, sfemFMD may provide direct information on endothelial function improvements following leg exercise training [37,38]. To our knowledge, this is the first study to evaluate sfemFMD responses to CIT supplementation alone and combined with RT.

Although CIT and ARG supplementations have increased plasma ARG levels, they were ineffective to increase FMD in healthy individuals [14]. In contrast, evidence indicates that ARG supplementation improves endothelial function in individuals with a baseline brachial FMD lower than 7% [39], suggesting benefits of ARG and CIT are more prominent in individuals with endothelial dysfunction. Previous literature has elucidated 4 weeks of CIT significantly increased brachial FMD compared to PL in hypertensive postmenopausal women [40]. Our study demonstrated for the first time that sfemFMD increased after CIT supplementation alone compared to PL. Similar to our study, Morita and colleagues [15] found that 800 mg/day of CIT supplementation improved brachial FMD after 4 weeks without a further improvement after 8 weeks in patients with vasospastic angina and endothelial dysfunction. Previous studies reported an increase in NO bioavailability via de novo ARG production after acute and short-term CIT supplementation in older adults with heart failure [41], middle-aged men [42], and obese postmenopausal women [43]. Moreover, acute dietary nitrate enhanced NO bioavailability and sfemFMD in healthy older men [44]. Taken together, improvements in leg endothelial function after 4 weeks of CIT supplementation may be due to an increased ARG bioavailability, resulting in NO-mediated vasodilation in the legs [45].

When SVLIRT was added to CIT supplementation for 4 weeks, we found that sfemFMD progressively increased compared with no change in PL. However, the increase in sfemFMD was not significant compared to 4 weeks. In contrast with our present finding, there is some evidence showing that RT can improve endothelial function. Beck and colleagues [46] observed 8 weeks of RT improved brachial FMD with improvements in plasma levels of NO and endothelin-1, a potent vasoconstrictor, in young pre-hypertensive adults. Relevant to the present study, brachial FMD improved after 12 weeks of leg SVLIRT compared with controls in normotensive postmenopausal women [31]. In addition, 8 weeks of CIT supplementation combined with WBVT (a strength training modality) increased NO levels in obese postmenopausal women [43]. The potential mechanism is an exercise-induced increase in shear stress that upregulates eNOS activity and enhances NO bioavailability [46] and FMD. Consequently, our findings suggest that 4 weeks of SVLIRT with PL did not improve sfemFMD due to the short-term intervention. However, CIT for additional 4 weeks plus SVLIRT was effective to improve leg endothelial function in hypertensive postmenopausal women.

There was a significant reduction in leg arterial stiffness (faPWV) by ~0.5 m/s after 4 weeks of SVLIRT in both groups. Consistent with our current finding, previous studies have shown that 8 and 12 weeks of WBVT reduced systemic (brachial–ankle PWV, baPWV) and leg (faPWV) arterial stiffness in hypertensive postmenopausal women [16,47]. This reduction in faPWV may be a result of increased NO bioavailability and decreased sympathetic modulation after SVLIRT, as demonstrated by Macedo and colleagues in rats [48]. The significant reduction in leg arterial stiffness is of clinical significance considering that each 1 m/s increase in baPWV is associated with a 12% augmented cardiovascular event risk [49], since the femoral–ankle arterial segment is the larger component of baPWV [50]. Thus, our findings suggest that 4 weeks of SVLIRT decreases leg arterial stiffness independently of CIT supplementation. A longer-term CIT plus SVLIRT intervention may be required to have an additional effect on arterial stiffness in postmenopausal women [16].

In older adults, reduced physical activity leads to declines in leg LM and anabolic resistance which is a critical factor to regain muscle mass with RT [23,51]. Age-related loss of muscle strength, especially in the legs, results in reduced physical performance and increased mortality risk [52,53]. Therefore, it is crucial to maintain leg muscle mass and strength in postmenopausal women [19]. The present study showed significant increases in leg LM after 8 weeks of CIT supplementation with the addition of SVLIRT during the last 4 weeks compared to PL. In older obese rats, 12 weeks of ARG supplementation alone enhanced soleus muscle mass via NO production and protein synthesis [54], but no effect was found in older adults [55]. On the other hand, findings in malnourished rats and older adults supplemented with CIT alone showed improvements in muscle protein synthesis [56] and total LM [21]. The current study did not find significant increases in leg LM after 4 weeks of CIT supplementation alone using a similar dose to the previous study, suggesting that CIT supplementation is not effective for muscle improvements in non-malnourished postmenopausal women. Regarding the training effect, combined WBVT and CIT supplementation for 8 weeks increased leg LM, an effect not observed after each intervention alone [16], suggesting an additive anabolic effect. In agreement with the present study, previous work revealed inefficiency of 8–12 weeks of SVLIRT alone for improving leg LM in postmenopausal women [16,32]. However, we are the first to report that CIT supplementation combined with SVLIRT improved LM in the trained (leg) muscles. Previously in older adults, Devries et al. [23] reported that leg LM was increased after 2 weeks of low-intensity RT without an additional effect of CIT (5 g/day). The discrepancy between the previous and our findings may be attributed to lower CIT dose and duration of the combined intervention. In addition, ARG (3 g/day) supplementation during a combined aerobic and strength training for 6 weeks did not increase LM in older adults [55]. Low ARG dose and high ARG catabolism by intestinal arginase may have resulted in reduced circulating ARG availability for effectiveness in skeletal muscles. In contrast, we found that 4 weeks of SVLIRT plus CIT is effective to enhance LM in postmenopausal women despite a short duration. Moinard et al. [57] proposed 10 g of CIT as the most appropriate dose for clinical use. Oral CIT was an efficient ARG precursor that results in an important increase in systemic ARG availability [57]. We recently reported that this CIT dose for 4 weeks improved brachial FMD by increasing ARG availability in postmenopausal women with hypertension [40]. Our findings suggest that 10 g daily of CIT supplementation may help to increase leg LM when combined with SVLIRT, by increasing leg endothelial function via an ARG/NO pathway-induced vasodilation [33,58]. Furthermore, our study found that the addition of 4-week SVLIRT to CIT reduced total percent fat which is a factor associated with developing endothelial dysfunction in older adults [5], suggesting the clinical importance. A similar dose of CIT for 3 weeks reduced total fat mass in older women [21], suggesting a lipolytic effect.

There was a significant increase in leg curl strength after 4 weeks of CIT supplementation alone. However, the increase in leg curl strength was greater after the combined intervention compared to PL. Similarly, Buckinx and colleagues [24] reported improvements in handgrip strength without increases in total LM following CIT supplementation (10 g/day) combined with high-intensity interval elliptical training for 12 weeks in dynapenic obese older adults. In addition, combined WBVT with PL or CIT supplementation for 8 weeks similarly increased leg press muscle strength in hypertensive postmenopausal women [16]. The current study also found similar improvements in leg muscle strength after only 4 weeks of SVLIRT with and without CIT, except for the leg curl. Our study showed a significant improvement in leg curl (hamstring) strength after CIT without SVLIRT, which progressively increased after the combined intervention. The benefit on the curl strength could be explained by lower muscle strength in hamstring than quadriceps in women [59]. This improvement may affect muscle balance between hamstring and quadriceps muscles, which is important to prevent slips and falls in older adults [60]. Of note, the previous studies were of longer durations (8 or 12 weeks) than the present, adding to the relevance of our findings. Endothelial dysfunction has been proposed as a mechanism of reduced muscle mass and strength in older women [5,6]. The possible mechanism is that CIT, as a precursor of de novo plasma ARG, enhances peripheral vasodilation via greater ARG-NO bioavailability [33,58,61] leading to greater delivery of nutrients, insulin, and oxygen to skeletal muscles [62]. Moreover, CIT supplementation directly stimulates muscle protein synthesis, which may contribute to improve muscle mass and strength in a malnourished state and sedentary aging [21,63]. Therefore, our findings suggest that vascular adaptations to CIT supplementation and SVLIRT may result in enhanced improvement in leg muscle mass and strength in hypertensive postmenopausal women.

This study has a few limitations. First, we included hypertensive postmenopausal women who were on anti-hypertensive medications and/or hormone replacement therapy. However, these women were well-controlled for a long time (taking medication >3 months) and were otherwise healthy. Second, although we asked participants not to change their daily activity and diet during the intervention period, we did not keep track of their compliance using food and activity logs. Third, we did not measure circulating ARG and NO levels, which would provide further explanation of potential mechanisms behind our findings. However, we directly assessed endothelial function using sfemFMD and found a significant improvement. We found that 8 weeks of CIT supplementation and 4 weeks of SVLIRT did not cause adverse effects in our participants, suggesting that the CIT dose and exercise training prescription were appropriate for postmenopausal women. Future studies are needed to investigate a long-term (≥8 weeks) effect of combined CIT supplementation with RT on vascular and muscular responses in various populations.

## 5. Conclusions

Our findings suggest that CIT supplementation improves leg endothelial function and has an additive effect on leg lean mass and curl strength when combined with SVLIRT. Therefore, CIT supplementation combined with SVLIRT is an effective therapeutic strategy to improve muscle mass and strength via enhanced endothelial function in postmenopausal women with hypertension.

## Figures and Tables

**Figure 1 nutrients-15-00074-f001:**
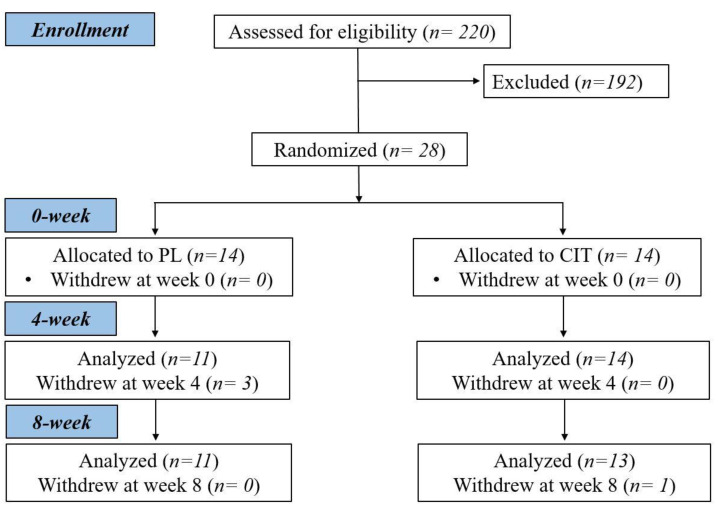
Study flow chart. CIT, L-citrulline; PL, placebo.

**Figure 2 nutrients-15-00074-f002:**
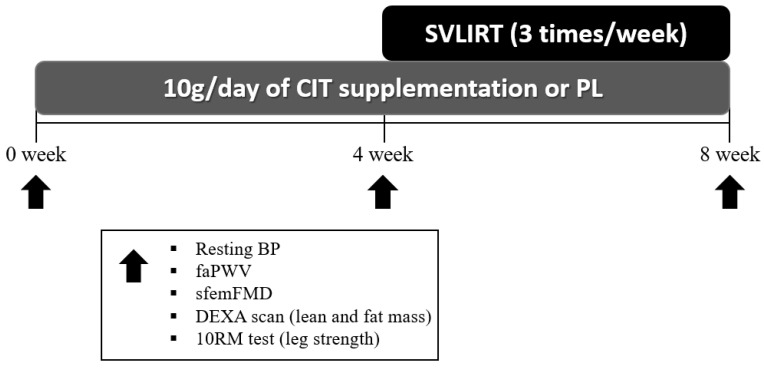
Experimental protocol. CIT, L-citrulline; PL, placebo; SVLIRT, slow-velocity low-intensity resistance training; BP, blood pressure; faPWV, femoral-ankle pulse wave velocity; sfemFMD, superficial femoral artery flow-mediated dilation; DEXA, whole-body dual-energy X-ray absorptiometry; RM, repetition maximum.

**Figure 3 nutrients-15-00074-f003:**
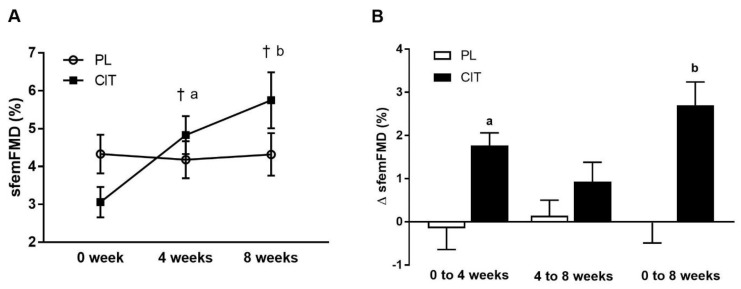
Superficial femoral artery flow-mediated dilation (sfemFMD) (**A**) at 0 week, after 4 weeks (CIT alone), and after 8 weeks (CIT and SVLIRT combined) and changes (Δ) in sfemFMD (**B**) from 0 week to 4 weeks, from 4 weeks to 8 weeks, and from 0 week to 8 weeks compared to PL in postmenopausal women. ^†^
*p* < 0.001 vs. 0 week; ^a^
*p* < 0.05 vs. PL; ^b^
*p* < 0.001 vs. PL.

**Figure 4 nutrients-15-00074-f004:**
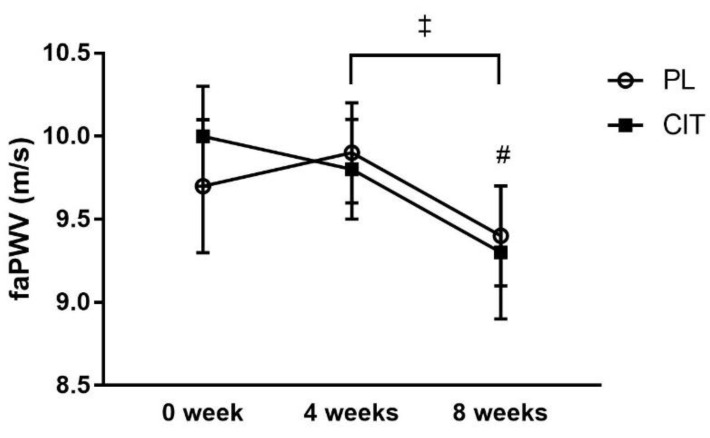
Effects of placebo (PL) and citrulline (CIT) supplementation on femoral-ankle pulse wave velocity (faPWV) after 4 weeks (CIT alone) and 8weeks (CIT and SVLIRT). ^#^
*p* < 0.05 vs. 4 weeks (for both groups); ^‡^
*p* < 0.001, time effect.

**Figure 5 nutrients-15-00074-f005:**
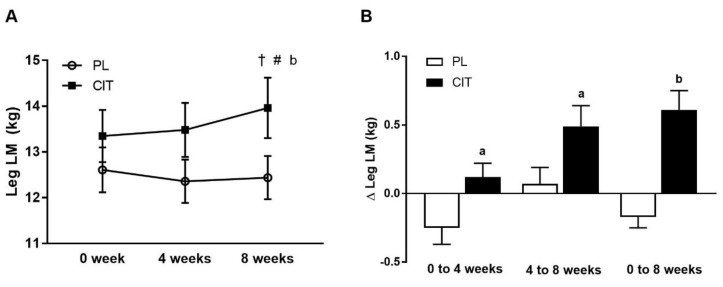
Legs lean mass (LM) (**A**) at 0 week, after 4 weeks (CIT supplementation alone), and after 8 weeks (CIT and SVLIRT combined) and changes (Δ) in legs LM (**B**) from 0 week to 4 weeks, from 4 weeks to 8 weeks, and from 0 week to 8 weeks compared to the PL in postmenopausal women. ^†^
*p* < 0.001 vs. 0 week; ^#^
*p* < 0.05 vs. 4 weeks; ^a^
*p* < 0.05 vs. PL; ^b^
*p* < 0.001 vs. PL.

**Figure 6 nutrients-15-00074-f006:**
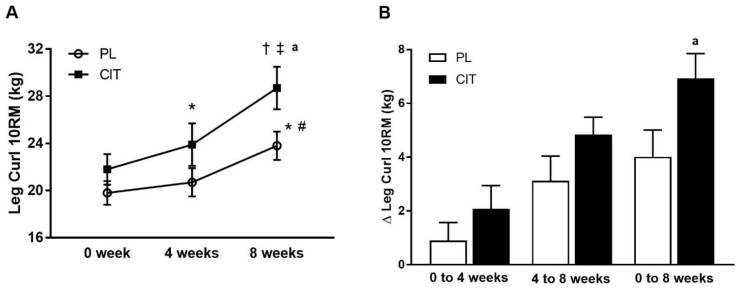
Leg curl 10-repetition maximum (10RM) (**A**) at 0 week, after 4 weeks (CIT supplementation alone), and after 8 weeks (CIT and SVLIRT combined) and changes (Δ) in leg curl 10RM (**B**) from 0 week to 4 weeks, from 4 weeks to 8 weeks, and from 0 week to 8 weeks compared to the PL in postmenopausal women. * *p* < 0.05 vs. 0 week; ^†^
*p* < 0.001 vs. 0 week; ^#^
*p* < 0.05 vs. 4 weeks; ^‡^
*p* < 0.001 vs. 4 weeks; ^a^
*p* < 0.05 vs. PL.

**Figure 7 nutrients-15-00074-f007:**
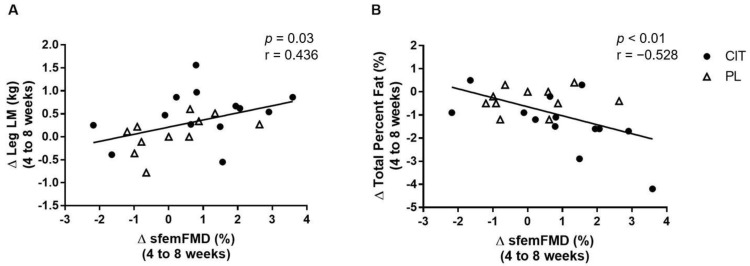
Correlations between changes (Δ) in superficial femoral artery flow-mediated dilation (sfemFMD) and leg lean mass (LM) (**A**) and total percent fat (**B**) from 4 weeks to 8 weeks.

**Table 1 nutrients-15-00074-t001:** Participant characteristics and medications.

Characteristics	PL (*n* = 11)	CIT (*n* = 13)	*p*
Age (years)	63 ± 1	62 ± 2	0.74
Height (m)	1.57 ± 0.02	1.56 ± 0.02	0.74
Weight (kg)	70.8 ± 3.8	72.3 ± 2.8	0.75
BMI (kg/m^2^)	29.2 ± 1.7	29.6 ± 1.1	0.85
Waist Circumference (cm)	93.6 ± 4.7	91.5 ± 2.9	0.70
Fasting Blood Glucose (mg/dL)	98.0 ± 4.7	95.0 ± 2.8	0.58
Resting SBP (mmHg)	135 ± 5	136 ± 5	0.85
Resting DBP (mmHg)	79 ± 3	77 ± 4	0.76
Resting MAP (mmHg)	98 ± 4	97 ± 4	0.89
**Hormone Replacement Therapy, *n***			
Estrogen	2	4	
Progesterone	1	0	
**Anti-hypertensive medications, *n***			
Diuretic	1	0	
ACE Inhibitor	2	2	
Ca^2+^ Channel Blocker	1	1	
ANG II Receptor Blocker	0	4	
Statins	1	1	
**Unmedicated, *n***	6	6	

Values are mean ± SE and *n*, number of subjects. Abbreviations: PL, placebo; CIT, Citrulline; BMI, body mass index; SBP, systolic blood pressure; DBP, diastolic blood pressure; MAP, mean arterial pressure; ACE, angiotensin converting enzyme; Ca^2+^, calcium; ANG II, angiotensin II. *p*-values are between-group differences from independent *t*-test.

**Table 2 nutrients-15-00074-t002:** Superficial femoral artery parameters and leg arterial stiffness at 0 (baseline), 4, and 8 weeks of placebo and citrulline supplementation alone (0–4 weeks) and combined with resistance training (4–8 weeks).

	PL (*n* = 11)			CIT (*n* = 13)			
Variables	0 week	4 weeks	8 weeks	0 week	4 weeks	8 weeks	*p*
Baseline diameter (mm)	6.1 ± 0.3	6.0 ± 0.3	6.0 ± 0.2	6.1 ± 0.2	5.9 ± 0.3	5.9 ± 0.2	0.35
Peak diameter (mm)	6.3 ± 0.3	6.2 ± 0.3	6.3 ± 0.2	6.3 ± 0.3	6.2 ± 0.3	6.2 ± 0.2	0.85
sfemFMD absolute (mm)	0.26 ± 0.03	0.23 ± 0.02	0.26 ± 0.04	0.19 ± 0.02	0.28 ± 0.03 ^†^	0.33 ± 0.04 ^†#^	<0.01
sfemFMD (%)	4.3 ± 0.5	4.2 ± 0.5	4.3 ± 0.6	3.1 ± 0.4	4.8 ± 0.5 ^†^	5.8 ± 0.7 ^†^	<0.001
Baseline shear rate (sec^−1^)	60 ± 10	78 ± 8	82 ± 12	66 ± 9	76 ± 8	84 ± 11	0.85
Peak shear rate (sec^−1^)	1546 ± 549	1834 ± 1002	395 ± 70	453 ± 505	1221 ± 923	513 ± 65	0.13
Shear rate AUC (au)	11567 ± 2678	13295 ± 2740	9493 ± 1903	9743 ± 2463	10062 ± 2521	9624 ± 1751	0.48
Shear rate AUC peak (au)	8376 ± 2119	9678 ± 2117	7616 ± 1488	6883 ± 1949	7552 ± 1947	7100 ± 1369	0.66
faPWV (m/s)	9.7 ± 0.4	9.9 ± 0.3	9.4 ± 0.3 ^#^	10.0 ± 0.3	9.8 ± 0.3	9.3 ± 0.4 ^#^	0.48

Values are mean ± SE. Abbreviations: PL, placebo; CIT, citrulline; sfemFMD, superficial femoral artery flow-mediated dilation; AUC, area under the curve; faPWV, femoral-ankle pulse wave velocity. *p*-values are group-by-time interaction from two-way repeated measure analysis of variance. ^†^
*p* < 0.001 vs. 0 week; ^#^
*p* < 0.05 vs. 4 weeks.

**Table 3 nutrients-15-00074-t003:** Body composition and leg strength variables at 0 (baseline), 4, and 8 weeks of placebo and citrulline supplementation alone (0–4 weeks) and combined with resistance training (4–8 weeks).

	PL (*n* = 11)			CIT (*n* = 13)			
	0 week	4 weeks	8 weeks	0 week	4 weeks	8 weeks	*p*
Height (m)	1.57 ± 0.02			1.56 ± 0.02			0.74
Weight (kg)	70.8 ± 3.8	72.2 ± 3.8	72.4 ± 3.8	72.3 ± 2.8	72.1 ± 2.9	72.5 ± 2.9	0.34
BMI (kg/m^2^)	29.2 ± 1.7	29.4 ± 1.7	29.4 ± 1.7	29.6 ± 1.1	29.5 ± 1.2	29.7 ± 1.2	0.76
WC (cm)	93.6 ± 4.7	93.7 ± 4.2	92.6 ± 4.6	91.5 ± 2.9	92.2 ± 2.7	90.8 ± 2.4	0.82
VAT (kg)	1.21 ± 0.23	1.20 ± 0.25	1.15 ± 0.26	0.84 ± 0.14	0.85 ± 0.14	0.81 ± 0.13	0.54
Total Percent Fat (%)	44.1 ± 2.0	44.6 ± 1.9	44.3 ± 2.0	42.9 ± 1.4	43.3 ± 1.0	42.0 ± 1.1 ^#^	0.27
Arms LM (kg)	4.14 ± 0.21	4.16 ± 0.23	4.15 ± 0.23	4.28 ± 0.22	4.31 ± 0.21	4.30 ± 0.22	0.91
Legs LM (kg)	12.6 ± 0.5	12.4 ± 0.5	12.4 ± 0.5	13.4 ± 0.6	13.5± 0.6	14.0 ± 0.7 ^†,#^	<0.001
Leg Press 10RM (kg)	88.1 ± 10.5	93.7 ± 10.2	100.2 ± 9.7 ^†,#^	96.4 ± 9.7	99.4 ± 9.3	109.6 ± 9.0 ^†,#^	0.76
Leg Extension 10RM (kg)	19.6 ± 1.5	22.2 ± 2.4	27.3 ± 2.3 *^,#^	20.9 ± 1.5	22.8 ± 1.8	27.7 ± 2.1 *^,‡^	0.77
Leg Curl 10RM (kg)	19.8 ± 1.0	20.7 ± 1.2	23.8 ± 1.2 *^,#^	21.8 ± 1.3	23.9 ± 1.8 *	28.7 ± 1.8 ^†,‡,a^	<0.05
Calf Raise 10RM (kg)	78.6 ± 5.9	82.6 ± 5.9	94.4 ± 5.1 *^,#^	84.4 ± 8.3	90.0 ± 9.5	102.6 ± 10.8 *^,#^	0.71

Values are mean ± SE. Abbreviations: PL, placebo; CIT, Citrulline; BMI, body mass index; WC, waist circumference; VAT, visceral adipose tissue; LM, lean mass; RM, repetition maximum. *p*-values in column are group-by-time interaction from two-way repeated measure analysis of variance. * *p* < 0.05 vs. 0 week; ^†^
*p* < 0.001 vs. 0 week; ^#^
*p* < 0.05 vs. 4 weeks; ^‡^
*p* < 0.001 vs. 4 weeks; ^a^
*p* < 0.05 vs. PL.

## Data Availability

The data presented in this study are available from the corresponding author upon request.

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
