# Peer review of "Combined L-Citrulline Supplementation and Slow Velocity Low-Intensity Resistance Training Improves Leg Endothelial Function, Lean Mass, and Strength in Hypertensive Postmenopausal Women"

_nutrients, 2022, doi:10.3390/nu15010074_

Round 1
Reviewer 1 Report
The manuscript is very detailed, well written and the topic is very current and interesting.
Author Response
We appreciate your comments.

Reviewer 2 Report
The article evaluated the role of L-Citrulline in hypertensive post-menopausal women. The idea is interesting because hypertension is a leading cause of endothelial dysfunction. Moreover, post-menopausal women suffers from a high-risk of cardiovascular diseases. In this scenario, the study disegn has some novelties. Nevertheless, I have some concerns to address. First, in Table 1, please add the comorbidities in baseline of the study population. Second, A multivariable analysis is highly recommended to confirm the results (adjusting for confounders). Third, regarding paragraph 2.5, why did you choose this kind of measurement, is it currently used in clinical practice? If not, have you been authorized in the protocol fom the Ethical Committe? Finally, I would suggest you to improve the discussion section speculating on the differences between L-Citrulline and L-Arginine, there you find some PMID to cite and discuss:
35339987
35498050
31336573
33562042
34836206
23075551
35868478
Author Response
First, in Table 1, please add the comorbidities in baseline of the study population.
Response: Our participants only had hypertension and were otherwise healthy. Thus, they did not have comorbidities.
Second, A multivariable analysis is highly recommended to confirm the results (adjusting for confounders).
Response: This was a small sample of postmenopausal women with hypertension, but otherwise healthy. We attempted to perform a multivariable analysis but since there were no associations between the significant improvements in sfemFMD, lean mass, or strength with participant’s characteristics, the authors found it more appropriate to report data using two-way repeated measures ANOVA.
Third, regarding paragraph 2.5, why did you choose this kind of measurement, is it currently used in clinical practice? If not, have you been authorized in the protocol from the Ethical Committee?
Response: Flow-mediated dilation (FMD) of the brachial artery is the non-invasive gold-standard test of endothelial function. Superficial femoral artery flow-mediated dilation (sfemFMD) is a widely measurement of endothelial function used in vascular research (PMID: 20952670). We used sfemFMD because our training protocol only used leg exercises. This measurement was approved by our IRB. The following articles (PMID) used sfemFMD as measure of endothelial function:
- 21512151, Thijssen et al. 2011. Journal of Applied Physiology, 111(1), 244-250.
- 18096601, Kooijman et al. 2008. The Journal of physiology, 586(4), 1137-1145.
- 31035478, Walker et al. 2019. Nutrients, 2019. 11(5): p. 954.
- 36079817, Smeets et al. 2022. Nutrients, 14(17): 3560.
Finally, I would suggest you to improve the discussion section speculating on the differences between L-Citrulline and L-Arginine, there you find some PMID to cite and discuss:
Response: Thank you for this suggestion. Please refer to lines 300-306, 351-353, 367-372, and 373-376 of the discussion section, which now discussed the difference between L-Citrulline and L-Arginine.
We considered these references (PMID):
- 19056561, Bai et al. 2009. The American journal of clinical nutrition, 89(1), 77-84.
- 96297080, Maharaj et al. 2022. Nutrients, 14(20), 4396.
- 19106310, Jobgen et al. 2009. The Journal of nutrition, 139(2), 230-237.
- 35337088, Córdova-Martínez A et al. 2022. Pharmaceuticals, 15(3), 290.
